# Anaphor Assisted Document-Level Relation Extraction

**Chonggang Lu[1], Richong Zhang[1,2]\*, Kai Sun[1], Jaein Kim[1],**
**Cunwang Zhang[1], Yongyi Mao[3]**

[1]SKLSDE, Beihang University, Beijing, China
[2]Zhongguancun Laboratory, Beijing, China
[3]School of Electrical Engineering and Computer Science, University of Ottawa, Ottawa, Canada

{lucg, zhangrc}@act.buaa.edu.cn, {sunkai,jaein,zhangcw}@buaa.edu.cn, ymao@uottawa.ca

## Abstract

Document-level relation extraction (DocRE) involves identifying relations between entities distributed in multiple sentences within a document. Existing methods focus on building a heterogeneous document graph to model the internal structure of an entity and the external interaction between entities. However, there are two drawbacks in existing methods. On one hand, anaphor plays an important role in reasoning to identify relations between entities but is ignored by these methods. On the other hand, these methods achieve cross-sentence entity interactions implicitly by utilizing a document or sentences as intermediate nodes. Such an approach has difficulties in learning fine-grained interactions between entities across different sentences, resulting in sub-optimal performance. To address these issues, we propose an Anaphor-Assisted (AA) framework for DocRE tasks. Experimental results on the widely-used datasets demonstrate that our model achieves a new state-of-the-art performance.[1]

## 1 Introduction

Document-level relation extraction (DocRE) has garnered increasing attention from researchers lately due to its alignment with real-world applications, where a large number of relational facts are expressed in multiple sentences (Yao et al., 2019). Compared with its sentence-level counterpart (Zhang et al., 2018; Zhu et al., 2019; Sun et al., 2020), DocRE is more difficult as it requires a more sophisticated understanding of context and needs to model the interaction in mentions belonging to the same or different entities distributed in multiple sentences (Yu et al., 2022; Xu et al., 2022, 2021).

To address DocRE, most of existing works use pre-trained language model, such as BERT (Devlin

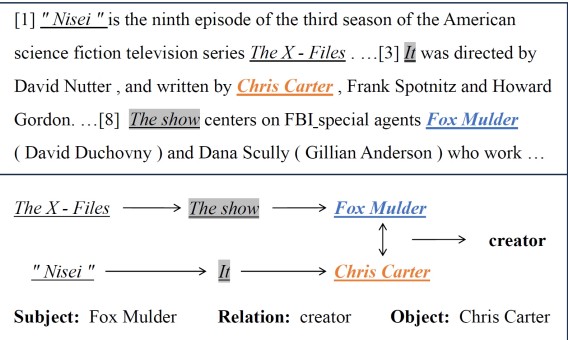

Figure 1: An example from DocRED dataset.

et al., 2019) and RoBERTa (Liu et al., 2019), to capture the long-range dependency between entities in the document, which are called the transformer-based models (Zhou et al., 2021; Zhang et al., 2021; Xiao et al., 2022a; Tan et al., 2022a). These models only take the word sequence as input without considering the internal structure of an entity and fail to explicitly learn the external interaction between entities, thus having difficulties in addressing complex instances that require reasoning (Zhou et al., 2021; Xie et al., 2022). To enhance the reasoning ability of transformer-based models, recent works propose to transfer the document into a graph where entity mentions are introduced as nodes and operate a graph neural network (Scarselli et al., 2008) on the document graph to explicitly learn information interactions between entities, which are usually called the graph-based models (Wang et al., 2020c; Zeng et al., 2020; Wang et al., 2020a).

Though some promising results have been achieved by recent graph-based models, cross-sentence entity interaction is not well learned in these models since most of them use a document node (Zeng et al., 2020) or sentence nodes (Wang et al., 2020a) as intermediate nodes to link entities in different sentences with an unclear information delivery. Such an approach captures interactions between entities distributed in different sentences

---

\*Corresponding author.
[1]Our code is available at https://github.com/BurgerBurgerBurger/AA.

via a common document node or sentence node, which is not effective in learning fine-grained interactions between entities, leading to sub-optimal performance. Besides, previous methods (Zhou et al., 2021; Zeng et al., 2020; Ma et al., 2023) are generally missing the anaphors in a document. However, identifying cross-sentence relations between entities often requires the intervention of anaphors since anaphors often convey the interaction between entities in multiple sentences as intermediate nodes, such as pronouns or definite referents, which refer to entities within the context.

As shown in Figure 1, to identify the relation *"creator"* between head entity *"Fox Mulder"* and tail entity *"Chirs Carter"*, sentences [1], [3], and [8] should be considered simultaneously. In these sentences, the anaphors play an important role in helping the model identify the target relation. Specifically, *"it"* in sentence [3] refers to *"Nisei"* in sentence [1] while *"The show"* in sentence [8] refers to *"The X-Files"* in sentence [1]. With these anaphors as bridges, the information transport between entities across sentences can be facilitated. For instance, the connection between *"Fox Mulder"* and *"The X-Files"* can be established via *"The show"*. Similarly, the connection between *"Nisei"* and *"Chirs Carter"* can be established via *"it"*. By leveraging the correspondence between these anaphors and entities, it is easier for the model to capture the semantic relations between *"Fox Mulder"* and *"The X-Files"*, *"Nisei"* and *"Chirs Carter"*, and finally promote extraction of the relation between *"Fox Mulder"* and *"Chirs Carter"*.

Based on these observations above, we are motivated to develop a new framework to explicitly and jointly leverage the coreference and anaphora information in the document. We achieve this by introducing a new document graph which is constructed by considering all possible entity mentions and anaphors in the document. To distinguish the importance of nodes and edges in the graph, we define three types of edges to connect the nodes and further propose an attention-based graph convolutional neural network to dynamically learn the structure of the graph. With the proposed framework, the information transport on the graph is sufficiently modeled and an expressive entity presentation is extracted for final classification. Following previous works (Ma et al., 2023; Xiao et al., 2022a; Xie et al., 2022), we also introduce evidence retrieval as an auxiliary task to help the model filter out irrelevant information. Extensive experiments on DocRED and Re-DocRED confirm the effectiveness of our proposed anaphor-assisted model. In summary, the main contributions of this paper are as follows:

- We propose a novel framework that explicitly and jointly models coreference and anaphora information, enabling the capture of fine-grained interactions between entities.

- We employ a dynamic algorithm for graph pruning and structure optimization, which necessitates minimal additional annotations.

- Experimental results show that our approach is valid and outperforms previous state-of-the-art document-level RE methods on two DocRE datasets.

## 2 Related Work

According to whether explicitly model the information interaction between entities, current works can be divided into two lines: transformer-based models and graph-based models.

### 2.1 Transformer-based Models

Transformer-based models take only the word sequence of a document as input and leverage the transformer (Vaswani et al., 2017) to implicitly capture the long-range contextual dependencies between entities. So far, lots of transformer-based models have been proposed for DocRE. A majority of these models focus on extracting more expressive entity representations from the output of transformer (Tan et al., 2022a; Xiao et al., 2022b; Zhou et al., 2021). Among these works, Zhou et al. (2021) propose a localized context pooling to enhance the representations of entities by locating the relevant context. Xie et al. (2022) propose an evidence-enhanced framework, EIDER, that effectively extracts and fuses evidence in the inference stage while Ma et al. (2023) uses evidence to construct supervision signals for model training with the aim to filter out irrelevant information. However, without considering the internal structure of an entity or explicitly learning the external interaction between entities, these transformer-based models are found to have difficulties in identifying relations in some complex instances that require reasoning (Zhou et al., 2021; Xie et al., 2022).

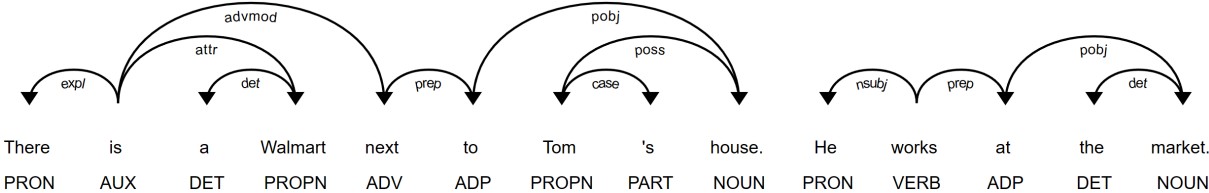

Figure 2: Dependency Tree and Part-of-Speech Tags for an example.

## 2.2 Graph-based Models

Graph-based models focus on building a document graph and explicitly learn the information interaction between entities based on the constructed graph. Most existing works define three types of nodes: mentions, entities, and sentences, and link nodes using heuristic rules, such as connecting mentions within the same entity or mentions within the same sentence but belonging to different entities (Li et al., 2020; Zhao et al., 2022; Sahu et al., 2019; Wang et al., 2020a; Zeng et al., 2020). For example, Christopoulou et al. (2019) introduces three types of nodes (mentions, entities, and sentences) to build the document graph and propose an edge-oriented graph neural network to operate over the constructed graph. Wang et al. (2020b) build a document graph similar to (Christopoulou et al., 2019) but design a global-to-local mechanism to encode coarse-grained and fine-grained semantic information of entities. Li et al. (2020) build a heterogeneous document graph that is comprised of entity nodes and sentence nodes with three types of edges: sentence-sentence edges, entity-entity edges, and entity-sentence edges. Zeng et al. (2020) propose to aggregate contextual information using a mention-level graph and develop a path reasoning mechanism to infer relations between entities.

However, these graph-based models focus on learning interactions between entities in the same sentence while interactions between entities in different sentences are not well modeled, thus having difficulties in handling the case shown in Figure 1. Besides, these models generally construct a static graph, assuming different edges are equally important. Although Nan et al. (2020) propose to treat the document graph as a latent variable and induce it based on attention mechanisms, their method ignores the co-referential information of entities and mentions, leading to sub-optimal performance. More importantly, previous methods generally overlook anaphors in the document, like *he*, *she*, *it*, and definite referents, which play an important role in cross-sentence relation extraction.

## 3 Problem Definition

Consider a document $D$ containing tokens $\mathcal{T}_D = \{t_i\}_{i=1}^{|\mathcal{T}_D|}$, sentences $\mathcal{S}_D = \{s_i\}_{i=1}^{|\mathcal{S}_D|}$, and entities $\mathcal{E}_D = \{e_i\}_{i=1}^{|\mathcal{E}_D|}$, the goal of document-level relation extraction is to predict the relation $r$ for each entity pair $(e_h, e_t)$ from a pre-defined relation set $\mathcal{R} \cup \{NA\}$, where $\{NA\}$ denotes no relation between two entities. Each entity $e \in \mathcal{E}_D$ is represented by its mentions $\mathcal{M}_e = \{m_i\}_{i=1}^{|\mathcal{M}_e|}$ and each mention $m \in \mathcal{M}_e$ is a phrase in the document. Except for the mentions, there are other phrases, known as *anaphors*, that may refer to the entities. Anaphors usually include pronouns like *he, she*, or *it*, and definite referents like *the song* or *the show*. Our purpose is to use the information of anaphors to promote relation extraction. Moreover, for each entity pair that has a valid relation $r \in \mathcal{R}$, a set of evidence sentences $\mathcal{V}_{h,t} \in \mathcal{S}_D$ is provided to specify the key sentences in the document for relation extraction.

## 4 Model

### 4.1 Anaphor Extraction

We use the off-the-shelf NLP tool, Spacy[2] to identify anaphors. Specifically, we employ Spacy's tagger functionality for Part-of-Speech tagging, treating all pronouns (PRON) as potential anaphors. Additionally, we utilize Spacy's dependency parser to aid in anaphor identification. In particular, when a token's dependency relation is 'det' (indicating determiner) and its text is 'the,' our approach involves identifying all tokens positioned between this specific 'the' token and its associated head as an anaphor. Figure 4.1 shows an example in the sentences 'There is a Walmart next to Tom's house. He works at the market.' In this example, 'He' and 'the market' are extracted as anaphors.

---

[2]https://spacy.io/

[1] _" Nisei "_ is the ninth episode of the third season of the American science fiction television series _The X - Files_ .

[3] _It_ was directed by David Nutter , and written by _**Chris Carter**_, Frank Spotnitz and Howard Gordon .

[8] _The show_ centers on _FBI_ special agents _**Fox Mulder**_ ( David Duchovny ) and Dana Scully ( Gillian Anderson ) _who_ …

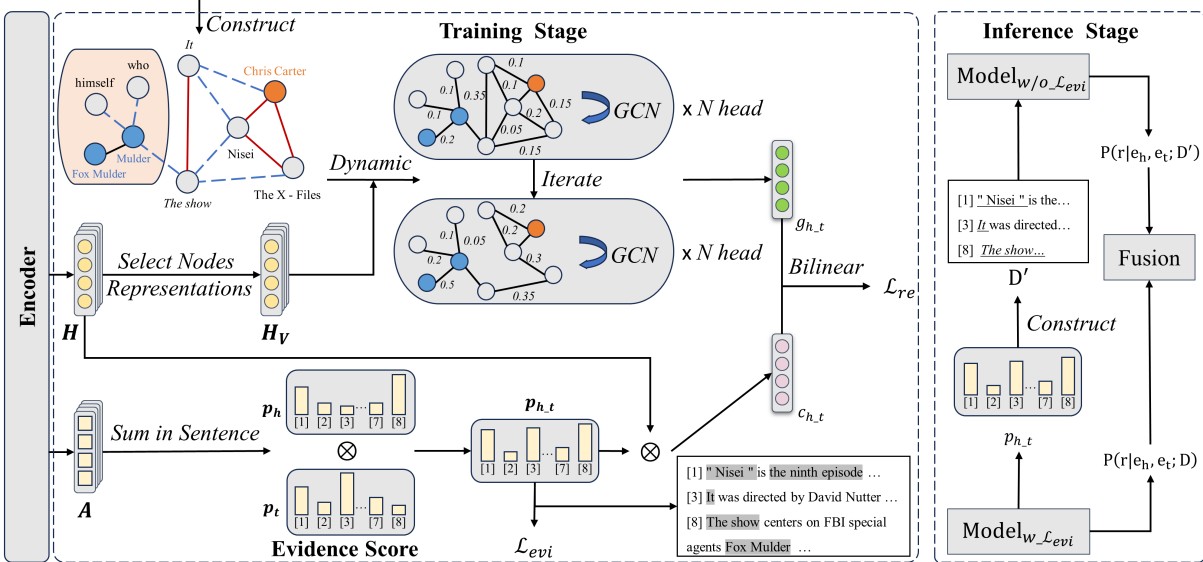

Figure 3: The overview of our anaphor-assisted DocRE framework.

## 4.2 Document Graph Construction

We transfer the document into an undirected graph $G = (V, E)$ which is constructed by considering all mentions and possible anaphors in the document. Specifically, we define two types of nodes: mention nodes and anaphor nodes. To distinguish the semantics of connections between different types of nodes, we introduce three types of edges: mention-anaphor edge, co-reference edge, and inter-entity edge. These edges represent anaphora resolution, coreference resolution, and external entity interaction respectively.

### 4.2.1 Graph Nodes

We collect two types of phrases in the document to build the node set $V$, mentions and anaphors. Mentions are annotated in the document while the anaphors are obtained by a semantic parsing tool.
**Mention Nodes:** Entities are represented by mentions in the document. In the graph, each mention is represented by a mention node. For example, in Figure 1, the mentions {_"Nisei"_, _Chris Carter_, _Fox Mulder_} are represented as mention nodes in the graph. Note that all mentions are annotated in datasets, and one entity may have multiple mentions.
**Anaphor Nodes:** Besides the mentions, there are other linguistic components that may refer to an entity. Specifically, in a document, there are pro-

nouns and definite referents that represent an entity. To exploit them, we create anaphor nodes for each pronoun and definite referent in the document. For example, in Figure 1, the pronoun _it_ and the definite referent _The show_ refer to _Nisei_ and _The X-Files_ and are represented as anaphor nodes in the graph.

### 4.2.2 Graph Edges

To this end, we obtained a node set $V$ composed of the mention and anaphora nodes. We then make connections between these nodes to obtain the edge set $E$. Specifically, there are three types of edges: mention-anaphor edges, co-reference edges, and inter-entity edges.
**Mention-Anaphor Edges:** First, anaphor nodes should be associated with their corresponding mention nodes and there should be mention-anaphor edges between them. However, the ground truth correspondence between mention and anaphor nodes is unknown. Thus, we consider all possible connections between the mention and the anaphor nodes. More specifically, we link each mention node with all anaphor nodes.
**Co-reference Edge:** Mentions that refer to the same entity are connected with each other. This allows modeling the internal structure of an entity as well as facilitating the information transport between multiple mention nodes of the entity to extract an expressive entity representation.

**Inter-Entity Edges:** To enhance information interaction between different entities in the document, we connect every pair of mention nodes that refer to different entities across the entire document. In other words, if two mentions refer to different entities, an edge exists between the corresponding mention nodes in the graph.

### 4.3 Document Encoding

Given a document $D$ containing $l$ tokens $\mathcal{T}_D = \{t_i\}_{i=1}^l$, we insert a special token "*" before and behind each mention as an entity marker (Zhang et al., 2017), with each mention $m_i$ represented by the embedding of "*" at the start position. For a pretrained language model (PLM) with a dimension of $d$, we feed the document $D$ to obtain the token embeddings $H$ and cross-token attention $A$:

$$H, A = \text{PLM}([t_1, t_2, ..., t_l]) \qquad (1)$$

where $H \in R^{l \times d}$ and $A \in R^{l \times l}$. Both $H$ and $A$ take advantage of the last three transformer layers of the PLM.

The embedding for each entity $e_i$ is obtained by applying *logsumexp* pooling (Jia et al., 2019) over embeddings of corresponding mentions. Formally, for each entity $e \in \mathcal{E}_D$, its representation $h_e$ is computed as follows:

$$h_e = \log \sum_{i=1}^{|M_e|} \exp(h_{m_i}) \qquad (2)$$

For each pair of entities $(e_h, e_t)$, the special contextual representation $c_{h,t}$ is obtained using the attention matrix $A$ and token embeddings $H$.

$$c_{h,t} = H^T \frac{A_h \otimes A_t}{A_h^T A_t} \qquad (3)$$

where $A_h \in R^l$ is attention to all the tokens for entity $e_h$, likewise for $A_t$. In other words, both $A_h$ and $A_t$ are vectors that represent the relevance of entities' tokens to all the tokens in the document. $\otimes$ denotes element-wise product.

### 4.4 Convolution over Dynamic Graph for Relation Extraction

We leverage the multi-head attention mechanism (Vaswani et al., 2017) to dynamically learn the graph structure to distinguish the importance of different edges. Specifically, the dynamic graph, containing n nodes, is represented by an $n \times n$ adjacency matrix $\tilde{A}$ with elements $\tilde{A}_{ij}$ in the range

of [0,1]. $\tilde{A}_{ij}$ represents the importance of the edge between two connected nodes $v_i$ and $v_j$, where a high $\tilde{A}_{ij}$ suggests a strong relationship between the two nodes otherwise weak. Formally, $\tilde{A}$ is computed as follows:

$$\tilde{A} = softmax(\sum_{u=1}^{|\mathcal{U}|} A_u \cdot \frac{H_V W_u^Q \times (H_V W_u^K)^T}{\sqrt{d}}), \qquad (4)$$

where $H_V$ is node representation , $\mathcal{U}$ denotes the set of edge types, $W_u^Q \in R^{d \times d}$ and $W_u^K \in R^{d \times d}$ are learnable parameter matrices, and $A_u$ is a binary adjacency matrix for edges of type $u$.

We then conduct graph convolutional networks (GCN) (Kipf and Welling, 2017) over the dynamic document graph to model the interaction between nodes. The information transport is guided by the learned $\tilde{A}$. This allows the information to be transferred between nodes along important paths identified by the learned weight $\tilde{A}_{ij}$. Let $g_i^{k-1}, g_i^k$ denote the input and output representation of $k^{\text{th}}$ GCN layer for node $v_i$, $g_i^k$ can be formally computed as follows:

$$g_i^k = \sigma(\sum_{j=1}^n \tilde{A}_{ij} W^k g_j^{k-1} + b^k) + g_j^{k-1} \qquad (5)$$

where $W^k$ and $b^k$ are learnable parameter matrix and bias, $\sigma$ is the Relu activation function.

After multiple convolutions on the document graph, we expect the mention nodes to have aggregated important information from relevant nodes. Similar to Eq. (2), we then apply *logsumexp* pooling on embeddings of mention nodes belonging to the same entity to obtain representations $g_h$ and $g_t$ for head and tail entities. To preserve the contextual information captured by the PLM, we concatenate entity representations induced from the outputs of PLM with $g_h$ or $g_t$ and employ the bilinear function to generate prediction probability of relation classification:

$$z_h = \tanh(W_h[h_{e_h} \| c_{h,t} \| g_h] + b) \qquad (6)$$

$$z_t = \tanh(W_t[h_{e_t} \| c_{h,t} \| g_t] + b) \qquad (7)$$

$$o = z_h^T W_r z_t + b_r \qquad (8)$$

where $h_{e_h}$ and $h_{e_t}$ are entity representations computed by Eq. (2), $c_{h,t}$ are contextual representation computed by Eq. (3), $W_h \in R^{d \times d}, W_t \in R^{d \times d}, W_r \in R^{d \times d}$ are learnable paramters.

## 4.5 Evidence Supervision Module

In the evidence supervision module, we introduce evidence retrieval as an auxiliary task and follow Ma et al. (2023) to utilize evidence distribution for enhancing the model's capability in filtering out irrelevant information. Specifically, for each entity pair $(e_h, e_t)$ with a valid relation, weights to all the tokens $q_{h,t} \in R^l$ is computed as:

$$q_{h,t} = \frac{A_h \otimes A_t}{A_h^T A_t} \qquad (9)$$

Then, for a sentence $s_i$ beginning from a token indexed by $m$ and ending at a token indexed by $n$, its weight is obtained by adding up all weights of tokens within $s_i$:

$$p_{h,t}^i = \sum_{j=m}^{n} q_{h,t}^j \qquad (10)$$

Let $p_{h,t} \in R^{|\mathcal{S}_D|}$ be the importance distribution for all sentences in the document. Then, we minimize the Kullback Leibler (KL) divergence between the extracted importance distribution $p_{h,t}$ and the evidence distribution $v_{h,t} \in R^{|\mathcal{S}_D|}$ derived from gold evidence labels:

$$\mathcal{L}_{evi} = -\sum_{h \neq t} v_{h,t} \ln \frac{v_{h,t}}{p_{h,t}}. \qquad (11)$$

## 4.6 Training Objective

Since there may be multiple relations between two entities, we formalize document-level relation extraction as a multi-label classification problem. Besides, a large portion of entity pairs have no valid relations. Following (Zhou et al., 2021), we introduce an adaptive threshold loss into our framework to address the relation imbalance issue. The loss of relation classification can be formalized as:

$$\mathcal{L}_{re} = -\sum_{r \in \mathcal{P}_T} \log\left(\frac{\exp(o_r)}{\sum_{r' \in \{\mathcal{P}_T, \text{TH}\}} \exp(o_{r'})}\right) \\ - \log\left(\frac{\exp(o_{\text{TH}})}{\sum_{r' \in \{\mathcal{N}_T, \text{TH}\}} \exp(o_{r'})}\right) \qquad (12)$$

where $\mathcal{P}$ denotes positive classes and $\mathcal{N}$ denotes negative classes for an entity pair $T = (e_h, e_t)$. TH denotes a threshold relation to differentiate between positive relation in $\mathcal{P}$ and negative relation in $\mathcal{N}$. This is achieved by adjusting the logits of the positive and negative relations such that the logits of positive relations are increased above the threshold value TH, while the logits of negative relations

are decreased below TH. This process enables the model to effectively distinguish between positive and negative relations in the input data.

We combine the loss of relation classification and evidence retrieval with a coefficient $\beta$. The total training loss of our model can be formalized as:

$$\mathcal{L} = \mathcal{L}_{re} + \beta \times \mathcal{L}_{evi}. \qquad (13)$$

## 4.7 Inference Stage Cross Fusion

Inference Stage Fusion (ISF) (Xie et al., 2022) was proposed to leverage RE predictions from the original document $D$ and pseudo document $D'$ constructed by evidence sentences. Two sets of prediction sores are merged through a blending layer (Wolpert, 1992) to obtain the final extraction results:

$$P_{fuse}(r|e_h, e_t) = P(r|e_h, e_t; D) \\ + P(r|e_h, e_t; D') - \tau \qquad (14)$$

where $\tau$ is a hyper-parameter tuned on development set.

ISF feeds the pseudo document into the model trained with evidence loss to obtain $P(r|e_h, e_t; D')$. The model trained with evidence loss naturally focuses on selecting the parts of interest from the input text, which is effective when the text contains a lot of irrelevant information. However, after the text has undergone one information filtering, i.e., evidence retrieval, the repeated information filtering process will lead to information loss, so we propose the Inference Stage Cross Fusion(ISCF) which uses a model trained without evidence loss to perform inference on the pseudo document $D'$.

## 5 Experiments and Analysis

### 5.1 Datasets

DocRED (Yao et al., 2019) is one of the most widely used datasets for document-level relation extraction. It contains 97 predefined relations, 63,427 relational facts, and 5,053 documents in total. However, Huang et al. (2022) have highlighted the considerable noise introduced by the recommend-revise annotation scheme employed to construct DocRED. To address the problem of missing labels within the DocRED, Tan et al. (2022b) proposed the Re-DocRED dataset which re-labels the DocRED dataset. Re-DocRED expands the quantity of relational facts in DocRED to a total of 119,991, while providing clean dev and test

| Datasets | DocRED | | Re-DocRED | | |
|---|---|---|---|---|---|
| | Train | Dev | Train | Dev | Test |
| #Docs | 3,053 | 1,000 | 3,053 | 500 | 500 |
| Avg. #Anaphors | 12.1 | 12.1 | 12.1 | 12.4 | 11.8 |
| Avg. #Mentions | 21.2 | 21.3 | 21.2 | 21.3 | 21.4 |
| Avg. #Entities | 19.5 | 19.6 | 19.4 | 19.4 | 19.6 |
| Avg. #Triples | 12.5 | 12.3 | 28.1 | 34.6 | 34.9 |
| Avg. #Sentences | 7.9 | 8.1 | 7.9 | 8.2 | 7.9 |

Table 1: Statistics of DocRED and Re-DocRED.

sets. The statistics of DocRED and Re-DocRED datasets, including anaphors and mentions, are presented in Table 1.

## 5.2 Implementation Details

Our model is implemented using the PyTorch library (Paszke et al., 2019) and HuggingFace Transformers (Wolf et al., 2019). All experiments are conducted on a single NVIDIA A100 40GB GPU. We employ BERT_base (Devlin et al., 2019) and RoBERTa_large (Liu et al., 2019) for DocRED and RoBERTa_large for Re-DocRED as document encoders. Num of GCN layers, attention heads, and iterates were all set to 2 in all experiments. All models are trained with the AdamW optimizer (Kingma and Ba, 2015), accompanied by a warm-up schedule to facilitate the training process. All hyper-parameters are tuned based on the dev set. We list some of the hyper-parameters in Table 2.

| Dataset | DocRED | | Re-DocRED |
|---|---|---|---|
| | BERT | RoBERTa | RoBERTa |
| epoch | 30 | 30 | 30 |
| lr_encoder | 5e-5 | 3e-5 | 3e-5 |
| lr_classifier | 1e-4 | 1e-4 | 1e-4 |
| batch size | 4 | 4 | 4 |
| warmup_ratio | 6e-2 | 6e-2 | 6e-2 |
| $\beta$ | 1e-1 | 3e-2 | 5e-2 |

Table 2: Best hyper-parameters of our model observed on the development set.

We employ F1, Ign-F1, Intra-F1, and Inter-F1 to evaluate the performance of our model. Ign-F1 measures F1 by disregarding relation triples present in the training set. Intra-F1 evaluates F1 for relation triples that do not require inter-sentence reasoning, whereas Inter-F1 evaluates F1 for relation triples that necessitate inter-sentence reasoning. To mitigate potential bias, we present the average results of our model across 5 independent runs with corresponding standard deviations. The results on the

DocRED test set were obtained by submitting the predictions to CodaLab[3].

## 5.3 Main Results

Our experimental results on DocRED are shown in Table 3, which indicates that our method consistently outperforms all strong baselines and existing SOTA model SAIS (Xiao et al., 2022a). Our BERT_base model exhibits notable enhancements in terms of F1 and Ign-F1, surpassing ATLOP-BERT_base by 1.8 and 1.53 on the test set. These improvements highlight the effectiveness of explicitly modeling the entity structure in the document. It is worth noting that our BERT_base model improves on F1 and Ign-F1 by 1.86 and 1.84 over the previous graph-based SOTA method GAIN-BERT_base, demonstrating the importance of using anaphors effectively to model cross-sentence entity interaction.

Table 4 presents a summary of the experimental results on Re-DocRED dataset. We observe a more pronounced performance gap between our model and baseline methods on Re-DocRED compared to the DocRED dataset. This disparity can be attributed to the much cleaner data annotations in the Re-DocRED dataset, ensuring a more fair basis for comparison. In Re-DocRED, our RoBERTa_large-based model surpasses ATLOP with F1 and Ign-F1 scores of 3.47 and 3.18 higher, respectively.

## 5.4 Ablation Study

To examine the effectiveness of different components in our model, we conduct a series of ablation studies on both DocRED and Re-DocRED, and the corresponding results are presented in Table 5. The detailed analysis is outlined below:

w/o Graph. Removing the dynamic graph leads to a degradation in model performance on both datasets, emphasizing the importance of explicitly modeling inner entity structure and cross-sentence entity interaction.

w/o ESM. We eliminate $\mathcal{L}\_{evi}$ in the training process and adopt a typical fusion process. The F1 score shows a decrease of 1.05, and 1.01 on the DocRED and Re-DocRED respectively. The decline in performance can be attributed to the factor of training without $\mathcal{L}\_{evi}$ which prevents the model from filtering out irrelevant information.

w/o ISCF w ISF. By replacing the cross-fusion in the inference stage with a typical fusion process,

[3]https://codalab.lisn.upsaclay.fr/competitions/365#results

| Model | PLM | Dev | | | | Test | |
|---|---|---|---|---|---|---|---|
| | | Ign-F1 | F1 | Intra-F1 | Inter-F1 | Ign-F1 | F1 |
| LSR (Nan et al., 2020) | BERT_base | 52.43 | 59.00 | 65.26 | 52.05 | 56.97 | 59.05 |
| ATLOP (Zhou et al., 2021) | BERT_base | 59.11[†] | 61.01[†] | 67.26[†] | 53.20[†] | 59.31 | 61.30 |
| GAIN (Zeng et al., 2020) | BERT_base | 59.14 | 61.22 | 67.10 | 53.90 | 59.00 | 61.24 |
| DocuNet (Zhang et al., 2021) | BERT_base | 59.86 | 61.83 | - | - | 59.93 | 61.86 |
| KD-DocRE (Tan et al., 2022a) | BERT_base | 60.08 | 62.03 | - | - | 60.04 | 62.08 |
| Eider (Xie et al., 2022) | BERT_base | 60.51 | 62.48 | 68.47 | 55.21 | 60.42 | 62.47 |
| DREEAM (Ma et al., 2023) | BERT_base | 60.51 | 62.55 | - | - | 60.03 | 62.49 |
| SAIS (Xiao et al., 2022a) | BERT_base | 59.98 | 62.96 | - | - | **60.96** | 62.77 |
| Ours | BERT_base | **61.31±0.07** | **63.38±0.08** | **69.41±0.14** | **55.92±0.22** | 60.84 | **63.10** |
| ATLOP (Zhou et al., 2021) | RoBERTa_large | 61.32 | 63.18 | 69.60 | 55.01 | 61.39 | 63.40 |
| DocuNet (Zhang et al., 2021) | RoBERTa_large | 62.23 | 64.12 | - | - | 62.39 | 64.55 |
| KD-DocRE (Tan et al., 2022a) | RoBERTa_large | 62.16 | 64.19 | - | - | 62.57 | 64.28 |
| Eider (Xie et al., 2022) | RoBERTa_large | 62.34 | 64.27 | 70.36 | 56.53 | 62.85 | 64.79 |
| DREEAM (Ma et al., 2023) | RoBERTa_large | 62.29 | 64.20 | - | - | 62.12 | 64.27 |
| SAIS (Xiao et al., 2022a) | RoBERTa_large | 62.23 | 65.17 | - | - | **63.44** | **65.11** |
| Ours | RoBERTa_large | **63.15 ± 0.05** | **65.19±0.09** | **71.09±0.08** | **57.83±0.13** | 62.88 | 64.98 |

Table 3: Performance comparison between our approach and previous SOTA baseline methods on DocRED dataset. Results with † are retrieved from Xie et al. (2022).

| Model | Dev | | Test | | | |
|---|---|---|---|---|---|---|
| | Ign-F1 | F1 | Ign-F1 | F1 | Intra-F1 | Inter-F1 |
| ATLOP (Zhou et al., 2021) | 76.88 | 77.63 | 76.94 | 77.73 | 80.18 | 75.13 |
| DocuNet (Zhang et al., 2021) | 77.53 | 78.16 | 77.27 | 77.92 | 79.91 | 76.64 |
| KD-DocRE (Tan et al., 2022a) | 77.92 | 78.65 | 77.63 | 78.35 | 79.57 | 77.26 |
| DREEAM (Ma et al., 2023) | - | - | 79.66 | 80.73 | - | - |
| PEMSCL (Guo et al., 2023) | 79.02 | 79.89 | 79.01 | 79.86 | - | - |
| Ours | **80.04±0.10** | **81.15±0.12** | **80.12±0.07** | **81.20±0.06** | **83.41±0.03** | **79.24±0.07** |

Table 4: Experimental results on Re-DocRED dataset. Results of existing methods are referred from Tan et al. (2022b) and their corresponding original papers. The reported results are all based on RoBERTa_large.

| Model | Ign-F1 | F1 | Intra-F1 | Inter-F1 |
|---|---|---|---|---|
| **DocRED** | | | | |
| Ours-BERT_base | **61.33** | **63.38** | 69.30 | **56.03** |
| $w/o$ Graph | 60.87 | 62.93 | 68.59 | 55.91 |
| $w/o$ ESM | 60.06 | 62.33 | 68.42 | 54.81 |
| $w/o$ ISCF $w$ ISF | 60.53 | 62.70 | 68.56 | 55.53 |
| $w/o$ Anaphor | 61.07 | 63.02 | **69.32** | 55.21 |
| Random replace | 60.96 | 63.01 | 69.28 | 55.30 |
| **Re-DocRED** | | | | |
| Ours-RoBERTa_large | **80.08** | **81.21** | 83.46 | **79.23** |
| $w/o$ Graph | 79.74 | 80.73 | 83.35 | 78.47 |
| $w/o$ ESM | 79.12 | 80.20 | 82.71 | 77.98 |
| $w/o$ ISCF $w$ ISF | 79.03 | 80.18 | 82.53 | 78.11 |
| $w/o$ Anaphor | 79.91 | 81.00 | 83.17 | 79.08 |
| Random replace | 79.70 | 80.89 | **83.47** | 78.67 |

Table 5: Ablation study on DocRED dev set and Re-DocRED test set. We use DocRED dev set due to the testing set of DocRED is not publicly available.

we observe a decrease in all metrics on two datasets. These results suggest that the duplicate information filtering process may not be justified to properly utilize pseudo-documents.

$w/o$ Anaphor or Random replace. Removing or replacing anaphors with randomly selected words resulted in a decrease in inter F1, while the intra F1 remained relatively consistent compared to scenarios with anaphor retention. This supports our motivation to use anaphors for enhancing cross-sentence interactions between entities.

## 5.5 Case Study

In Figure 4, a case study of our method is presented, where we examine three sentences within a document containing mentions such as *Carol II, Romania, Carol I* and *Zizi Lambrino*. Both ATLOP and DocuNet encounter difficulties in predicting the relationship of *spouse* between *Carol II* and *Zizi Lambrino*, primarily due to their failure in recognizing crucial anaphors, such as *he, his*. In contrast, our model adeptly captures anaphora information within the sentences, facilitating a more comprehensive understanding of the interaction between *Carol II* and *Zizi Lambrino*, ultimately leading to

[0] *Carol II* ( 15 October 18934 April 1953 ) reigned as King of *Romania* from 8 June 1930 until his coerced…
[1] *Carol* was the eldest son of Ferdinand I and became crown prince upon the death of his grand – uncle, King *Carol I* in 1914.
[4] He possessed a hedonistic personality that contributed to the controversies marring his reign , and his life was marked by numerous scandals , among them marriages to *Zizi Lambrino* and….

Figure 4: A case study of our method. Only a part of entities and sentences are displayed due to space limitation.

the successful identification of their relationship.

## 5.6 Impact of GCN Layers $K$

We conducted experiments on DocRED based on BERT_base without fusion to analyze the potential influence of the number of GCN layers. As depicted in Figure 5, there is a significant increase in the Intra-F1 score when transitioning from 0 to 1 GCN layer. This can be attributed to the reason that GCN facilitates efficient information transfer among different nodes such as $m_i \leftarrow it$ and $m_i \leftarrow m_j$. But a single layer of GCN is still insufficient to explicitly capture cross-sentence entity interactions such as $m_i \rightarrow the\ show \rightarrow m_j$ or $m_i \rightarrow it \rightarrow m_j$. The Inter-F1 score increase when the number of GCN layers is set to 2. Two-layer GCNs can enhance cross-sentence entity interactions by effectively aggregating information through anaphors.

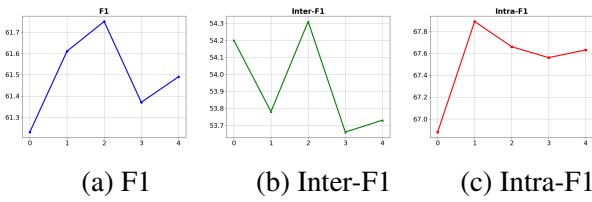

(a) F1      (b) Inter-F1      (c) Intra-F1

Figure 5: Impact of GCN layers $K$.

## 5.7 Impact of Coefficient $\beta$

In our proposed method, the evidence assistance coefficient $\beta$ plays a crucial role in regulating the trade-off between evidence retrieval loss and relation classification loss. In Figure 6, as the value of $\beta$ increases, the F1 score shows an overall trend of initially ascending and then subsequently declining. This pattern suggests that the optimal balance between $\mathcal{L}_{re}$ and $\mathcal{L}_{evi}$ tends to fall within the range of 0 to 0.1. It is evident that the choice of $\beta$ greatly affects the effectiveness and efficacy of the RE task at the document level.

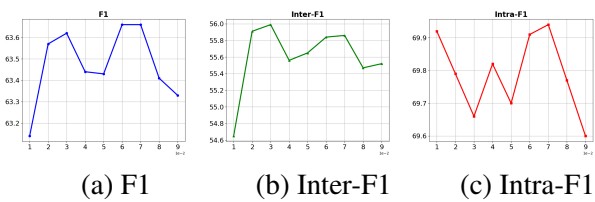

(a) F1      (b) Inter-F1      (c) Intra-F1

Figure 6: Impact of evidence assistance coefficient $\beta$.

## 6 Conclusions

In this paper, we present an approach that explicitly and jointly models coreference and anaphora information to effectively capture entities' internal structure and external interactions. Moreover, we employ a dynamic algorithm for graph pruning and structural optimization, which requires minimal additional annotations. We also introduce evidence retrieval as an auxiliary task to enhance the encoder. Empirical studies conducted on well-established benchmarks confirm the effectiveness of our proposed model.

## Acknowledgements

This work is supported partly by the National Key R&D Program of China under Grant 2021ZD0110700, partly by the Fundamental Research Funds for the Central Universities, and partly by the State Key Laboratory of Software Development Environment.

## Limitations

Our method has certain limitations that should be acknowledged. Firstly, the anaphors used in our method are acquired by an external parser, which has a risk of introducing potential errors caused by the external parser. Secondly, our model's generalization may be insufficient in cases where the document contains a limited number of anaphors.

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
