# OpenReview forum: "Anaphor Assisted Document-Level Relation Extraction"
_EMNLP/2023/Conference — EMNLP 2023 Main_

### Official Review · Reviewer_cmU3 · 2023-08-01

**Soundness:** 3

**Excitement:**

3: Ambivalent: It has merits (e.g., it reports state-of-the-art results, the idea is nice), but there are key weaknesses (e.g., it describes incremental work), and it can significantly benefit from another round of revision. However, I won't object to accepting it if my co-reviewers champion it.

**Paper Topic And Main Contributions:**

The paper highlights that inadequate preprocessing of the anaphor has hindered the ability to comprehend intricate interactions between entities. In response, the authors propose an anaphor-assisted framework for the DocRE task. The experiments demonstrate the efficacy of the proposed approach.

**Reasons To Accept:**

1. The proposed methods exhibit commendable performance.
2. The paper is lucid and skillfully composed.

**Reasons To Reject:**

1. The paper utilise mutlple teisting techniques and does not highlight the proposed methods.
2. The disparity between the motavtion and the experiments analysis. The paper proposed to explicitly and jointly model coreference and anaphora information and utilize a dynamic algorithm. However the experiments did not focus on the analysis of the proposed graph structure related to the anaphora, rather than the existing techniques.
3. The experiments are not completion. To solve the problem of anaphora, the paper proposes to introduce the anaphora node. To show the effective of the graph structure, the ablation of the anaphora node should be include.

**Reproducibility:**

4: Could mostly reproduce the results, but there may be some variation because of sample variance or minor variations in their interpretation of the protocol or method.

**Reviewer Confidence:**

4: Quite sure. I tried to check the important points carefully. It's unlikely, though conceivable, that I missed something that should affect my ratings.

---

> ### Author Rebuttal · Authors · 2023-08-29
>
> **Reasons To Reject:**
>
> R1:
>
> Thank you for your comment. In our view, we believe the utilization of multiple testing techniques serves to validate the robustness and effectiveness of our proposed approach. Regarding the lack of highlighting the proposed method, we intend to address this concern by including a summary in the introduction to highlight the contributions of our work, allowing readers to quickly grasp the novel aspects of the paper.
>
>
>
> R2:
>
> We appreciate the suggestion regarding the need for an analysis of the proposed graph structure in relation to the anaphora. In response, we have conducted two distinct experiments. The first involves removing anaphora nodes from the document graph (w/o anaphora), while the second replaces anaphora nodes with randomly selected words from the sentences (Random replace). The results are presented as following:
>
> | **Model**              | **Ign-F1** | **F1** | **Intra-F1** | **Inter-F1** |
> | ---------------------- | ---------- | ------ | ------------ | ------------ |
> | DocRED_BERT_base       | 61.33      | 63.38  | 69.30        | 56.03        |
> | w/o anaphor            | 61.07      | 63.02  | 69.32        | 55.21        |
> | Random replace         | 60.96      | 63.01  | 69.28        | 55.30        |
> | ReDocRED_RoBERTa_large | 80.08      | 81.21  | 83.46        | 79.23        |
> | w/o anaphor            | 79.91      | 81.00  | 83.17        | 79.08        |
> | Random replace         | 79.70      | 80.89  | 83.47        | 78.67        |
>
> The key insight drawn from this experiment is that the results on both the DocRED and ReDocRED datasets highlight the importance of retaining anaphora nodes in the proposed graph structure. We will include these results in our revised version.
>
>
>
> R3:
>
> Your suggestion to include an ablation study involving the anaphora node is insightful, and we will include the ablated results of removing anaphora nodes in the revised version.
>
> As the results in the table shown in R2, the removal of anaphora nodes was observed to lead to a decrease in inter F1, while the intra F1 remained relatively consistent compared to scenarios where anaphora nodes were retained. This aligns well with our motivation, which centers on utilizing anaphora to enhance cross-sentence interactions between entities.

---

### Official Review · Reviewer_VsBi · 2023-08-04

**Typos Grammar Style And Presentation Improvements:** 1. The equation reference in the pape…
**Soundness:** 4

**Excitement:**

4: Strong: This paper deepens the understanding of some phenomenon or lowers the barriers to an existing research direction.

**Paper Topic And Main Contributions:**

In this paper, the authors proposed an anaphor-assisted framework for DocRE tasks.  In order to explicitly and jointly leverage the co-reference and anaphora information in the document, the authors constructed a  document graph by considering all possible entity mentions and anaphors. They first employed an attention-based graph convolutional neural network to learn the structure of the graph dynamically. Additionally, they introduced evidence verification as an auxiliary task to assist the model in filtering out irrelevant information.

**Reasons To Accept:**

1. The experiment is relatively detailed, and the logic of the article is relatively easy to understand.
2. Experimental results on the widely-used datasets demonstrate that the model achieves a new state-of-the-art performance.

**Reasons To Reject:**

 1. In the introduction section, there is a lack of a summary regarding the innovations of this paper.
2. In Figure 5, it is not clear how the value of the evidence assistance coefficient β affects the model performance.

**Reproducibility:**

5: Could easily reproduce the results.

**Reviewer Confidence:**

1: Not my area, or paper was hard for me to understand. My evaluation is just an educated guess.

---

> ### Author Rebuttal · Authors · 2023-08-29
>
> **Reasons To Reject:**
>
> R1:
>
> We appreciate your feedback regarding the introduction section. Following your suggestion, we will provide a summary of our contributions to ensure that readers can quickly grasp the novel aspects of the paper.
>
>
>
> R2:
>
> In our paper, we have incorporated a multi-task learning setup, which involves optimizing multiple loss functions simultaneously (Eq.(13)). The evidence assistance coefficient β enables us to regulate the emphasis placed on learning from the two tasks simultaneously. In Figure 5, as the value of β increases, the F1 score shows an overall trend of initially ascending and then subsequently declining. This pattern suggests that the optimal balance between loss_re and loss_evi tends to fall within the range of 0 to 0.1.
>
> In the revised version, we will provide a more detailed explanation to highlight how varying the value of β leads to changes in the F1 score, with an effort to improve the clarity of this experiment.
>
> **Typos Grammar Style And Presentation Improvements:**
>
> R1:
>
> Thank you for pointing out the equation reference error on line 334. We apologize for this oversight and will make the necessary corrections in the revised version.

---

### Official Review · Reviewer_dWHw · 2023-08-09

**Soundness:** 4

**Excitement:**

4: Strong: This paper deepens the understanding of some phenomenon or lowers the barriers to an existing research direction.

**Paper Topic And Main Contributions:**

This work prosed an anaphor-assisted framework for document-level relation extraction. This framework leverages pronouns to capture the fine-grained interactions between entities.

**Questions For The Authors:**

1. May I know how are the anaphor nodes obtained? On average, how many anaphors and mentions are there in a document?
2. Besides, suppose the document has M mentions and N anaphors on average, does that mean there are M*N Mention-Anaphora Edges? or there are pruning mechanisms for such edges?

**Reasons To Accept:**

To the best of my knowledge, this is the first work to integrate pronoun-level coreference information in document-level RE, which is important for cross-sentence relation discovery. This work also leverages GNNs to learn Mention-Anaphora edges, which require minimal extra annotations. This framework improves the performances on DocRED and Re-DocRED datasets.

**Reasons To Reject:**

The presentation could be better. GNN systems are already proposed in document-level RE. In my opinion, the main contribution of the paper is introducing anaphor nodes and edges. However, the details on how to obtain such anaphors are not provided in section 4.1.

**Reproducibility:**

4: Could mostly reproduce the results, but there may be some variation because of sample variance or minor variations in their interpretation of the protocol or method.

**Reviewer Confidence:**

4: Quite sure. I tried to check the important points carefully. It's unlikely, though conceivable, that I missed something that should affect my ratings.

**Typos Grammar Style And Presentation Improvements:**

It will be better if the authors could provide more details on the method that they used to obtain the Anaphora Nodes. The concept of Anaphora Nodes is important.

---

> ### Author Rebuttal · Authors · 2023-08-29
>
> **Questions For The Authors:**
>
> R1:
>
> Thank you for bringing this to our attention. Regarding the details on obtaining the anaphor nodes, we wish to provide a detailed description here.
>
> Our approach to obtaining anaphor nodes involves two distinct rules (Lines 443 to 446):
>
> 1. We utilize Spacy's tagger functionality to perform parts-of-speech tagging, treating all pronouns (PRON) as potential anaphors, as they may refer back to a mention.
> 2. We also leverage Spacy's dependency parser to help identify anaphors. Specifically, in instances where a token's dependency relation is identified as 'det' (i.e., determiner) and its text is 'the', the approach involves identifying all tokens positioned between this particular 'the' token and its associated head as an anaphor. For instance, if the associated head of the ‘the’ token happens to be the word ‘song’ in the text ‘The song is a rock with elements of alternative rock.’, the resulting anaphor extracted is ‘the song’.
>
> Addressing your inquiry regarding the number of anaphors and mentions in a document on average, we provide the statistics here:
>
> |                | DocRED_train | DocRED_**dev** | DocRED_**test** | **ReDocRED_train** | **ReDocRED_dev** | ReDocRED_**test** |
> | -------------- | ------------ | -------------- | --------------- | ------------------ | ---------------- | ----------------- |
> | #Anaphors      | 36990        | 12088          | 11964           | 36990              | 6189             | 5899              |
> | Avg. #Anaphors | 12.1         | 12.1           | 12.0            | 12.1               | 12.4             | 11.8              |
> | #Mentions      | 64876        | 21336          | 21468           | 64876              | 10652            | 10684             |
> | Avg. #Mentions | 21.2         | 21.3           | 21.5            | 21.2               | 21.3             | 21.4              |
>
> We will include these additional details to further strengthen our work.
>
>
>
> R2:
>
> Indeed, if a document holds an average of M mentions and N anaphors, it implies the presence of M*N Mention-Anaphora Edges (Lines 247 to 255) within our constructed graph. Our model depends on an attention mechanism (Eq.(4))  as a soft pruning mechanism to dynamically select important Mention-Anaphora Edges in the graph.
>
>
>
> **Typos Grammar Style And Presentation Improvements:**
>
> R1:
>
> Thank you for your suggestion regarding further details about the method used to obtain Anaphora Nodes. In the revised version of our paper, we will provide additional details to ensure a clearer understanding of this aspect.

---

### Meta-Review · Area_Chair_pvUj · 2023-09-14

**Recommendation:** 4

**Metareview:**

This paper is well-written, and propose an innovative method for document-level relation extraction. Presentation should be improved according to reviewers’ comments.

---

### Decision · Program_Chairs · 2023-10-07

**Decision:**

Accept-Main

**Comment:**

This paper is well-written, and propose an innovative method for document-level relation extraction. Presentation should be improved according to reviewers’ comments.